# Eating Behavior and Obesity in a Sample of Spanish Schoolchildren

**DOI:** 10.3390/ijerph20054186

**Published:** 2023-02-26

**Authors:** Andrea Calderón García, Ana Alaminos-Torres, Roberto Pedrero Tomé, Consuelo Prado Martínez, Jesús Román Martínez Álvarez, Antonio Villarino Marín, María Dolores Marrodán Serrano

**Affiliations:** 1Research Group EPINUT (Nutritional Epidemiology), Faculty of Medicine, Complutense University of Madrid, 28040 Madrid, Spain; 2Department of Nursing and Nutrition, Faculty of Biomedical Sciences, Universidad Europea de Madrid, Villaviciosa de Odón, 28670 Madrid, Spain; 3Department of Biology, Faculty of Sciences, Universidad Autónoma de Madrid, 28049 Madrid, Spain; 4Department of Biodiversity, Ecology and Evolution, Faculty of Biology, Complutense University of Madrid, 28040 Madrid, Spain

**Keywords:** appetite, satiety response, eating behavior, pediatric obesity

## Abstract

From the point of view of prevention, it is convenient to explore the association between eating behavior and the obese phenotype during school and adolescent age. The aim of the present study was to identify eating behavior patterns associated with nutritional status in Spanish schoolchildren. A cross-sectional study of 283 boys and girls (aged 6 to 16 years) was carried out. The sample was evaluated anthropometrically by Body Mass Index (BMI), waist-to-height ratio (WHtR) and body fat percentage (%BF). Eating behavior was analyzed using the CEBQ “Children’s Eating Behavior Questionnaire”. The subscales of the CEBQ were significantly associated with BMI, WHtR and %BF. Pro-intake subscales (enjoyment of food, food responsiveness, emotional overeating, desire for drinks) were positively related to excess weight by BMI (β = 0.812 to 0.869; *p =* 0.002 to <0.001), abdominal obesity (β = 0.543–0.640; *p* = 0.02 to <0.009) and high adiposity (β = 0.508 to 0.595; *p* = 0.037 to 0.01). Anti-intake subscales (satiety responsiveness, slowness in eating, food fussiness) were negatively related to BMI (β = −0.661 to −0.719; *p* = 0.009 to 0.006) and % BF (β = −0.17 to −0.46; *p* = 0.042 to *p* = 0.016).

## 1. Introduction

Eating habits acquired during childhood and adolescence tend to become established during adulthood. For this reason, achieving a healthy diet at an early age is a definite factor in avoiding obesity and chronic diseases. Several experimental studies and reviews on the subject have shown that parents have a strong influence on their children’s eating behavior [1,2]. This is particularly important in childhood, during which children learn what, when and how to eat according to the cultural transmission of family patterns and attitudes [3,4]. Parental prohibition or restriction of food, or the use of food as a reward, are factors that impact the emotional domain and predict children’s enjoyment of food or their response to satiety [5]. Similarly, healthy nutrition education by families is associated with positive attitudes towards food and appropriate regulation of food intake which is reflected in children’s improved nutritional status [6]. Obviously, parents also pass on their genes, which also play a proven role in the regulation of appetite and food preferences [7,8,9]. In any case, eating behavior, which undoubtedly has a genetic and environmental component, is reflected in the nutritional condition of the subject and modulates the risk of obesity. 

Different studies conclude that the capacity to respond to satiety is lower in overweight children and adolescents, especially in those who are obese, as well as a more noteworthy response to food cues. They have understood this as a higher desire to eat and greater likelihood of ingestion in the presence of food. For this reason, overweight children and adolescents seem to be more likely to eat food in the absence of hunger, out of mere desire or pleasure [10]. In addition, food enjoyment and speed of intake appear to be higher in obese children, who have a delayed sense of satiety [11]. Therefore, this bidirectional association leads to children with a greater enjoyment or taste for food being at greater risk of obesity [12]. It is worth noting that a greater increase in intake under emotional stress has also been observed in overweight children and adolescents compared to medium and underweight subjects [13,14]. However, the results in this aspect are controversial as recent meta-analysis studies show that the relationship between emotional intake and body composition is not as direct in children and adolescents as in adults [15]. Consequently, it is necessary to explore the association between eating behavior and the obese phenotype during the school and adolescent age range.

Previous findings show the usefulness of analyzing the eating behavior of children in detail using questionnaires such as the Children’s Eating Behavior Questionnaire (CEBQ) [16]. This test identifies different phenotypes related to habits such as food avoidance, early or late satiety, gluttony, or tendency for emotional overeating, habits that may eventually alter nutritional status [17,18]. Research using the CEBQ relates overweight and obesity in children and adolescents with higher scores on the pro-intake scales and lower scores on the anti-intake scales, pointing to higher consumption and enjoyment of food, lower satiety and more emotional overeating behaviors. Conversely, low weight is associated with lower scores on the pro-intake scales and higher scores on the anti-intake scales, relating to avoidance eating behaviors, early satiety and lower enjoyment of food [19]. 

Initially used in British children [16], the CEBQ has been applied to schoolchildren from different populations, such as the United States [20], Sweden [21], Saudi Arabia [22], Bosnia [23], Portugal [24] and Chile [25]. In Spain, the only precedent is the study of Jimeno Martinez et al. [26] as part of the MELI-POP (Mediterranean Lifestyle in Pediatric Obesity Prevention) pilot study. On the other hand, in most of the mentioned studies, the association between eating behavior assessed by CEBQ and obesity has been established through weight and BMI, with very few studies that include other indicators of adiposity [27]. For this reason, the main objective of the present study is to identify, in a sample of Spanish schoolchildren, the eating behavior associated with nutritional status assessed by anthropometric parameters that identify, in more outstanding detail, body composition and fat distribution.

## 2. Materials and Methods

### 2.1. Participants 

This is a cross-sectional study in a convenience sample of 283 Spanish schoolchildren aged 6 to 16 years (33.21% (94) girls); (66.69% (189) boys). A total of 54.6% were aged between 6 and 10 years (107 boys and 48 girls). The remaining 45.40% (84 boys and 44 girls) were between 11 and 16 years of age. The sample was recruited between 2019 and 2021 in public schools and municipal sports centers in middle-class neighborhoods in the Community of Madrid, Spain. In these sports centers, schoolchildren perform soccer, basketball, gymnastics, or swimming activities as part of after-school classes. 

In 42.20% of families, both parents had primary education. In 25.30%, at least one parent had secondary or university education and in 32.50% of the cases, both parents had advanced specific vocational training or university education. All the schoolchildren performed between 100 and 120 min of physical activity per week during school hours in two sessions. A total of 93.20% also participated in out-of-school physical activity (mean = 3.61 SD = 1.84 h/week) with no differences between sexes (Table A1).

Data collection was carried out as part of a school health program developed by the Spanish Society of Dietetics and Food Sciences in coordination with local councils. It should be noted that data collection was partially affected by the COVID 19 pandemic, which forced special precautions and decreased the potential number of children finally included in the present study. The data were anonymized and were disaggregated from information that could identify the subject. Participants’ assent and informed consent from parents or guardians were required following the bioethical principles of the Declaration of Helsinki in its most updated version [28]. The Ethics Committee approved the project of the Autonomous University of Madrid (CEI-91-1699).

### 2.2. Instruments

Each participant was assessed anthropometrically through direct measurements, body composition indicators and adiposity distribution. Their parents or guardians completed the CEBQ [16] questionnaire. 

#### 2.2.1. Anthropometric Study

The anthropometric assessment was carried out according to the protocol of the International Biological Program (IBP) [29]. Height (cm) was measured with a Tanita Leicester measuring rod with an accuracy of 1 mm; weight (kg), umbilical waist circumference (cm) with a Cescorf tape and bicipital, tricipital, subscapular and suprailiac skinfolds (mm) with a Holtain adipometer with an accuracy of 0.2 mm and constant pressure (10 g/mm^2^). 

For prevalence analysis, the sample was stratified by sex. Nutritional categories were established based on the Body Mass Index [BMI = weight (kg)/height (m^2^)] using the cut-off points of Cole et al. [30,31] and the waist-to-height ratio (WHtR = waist circumference/height), using the criteria established by Marrodán et al. [32] which define abdominal obesity as >0.51 in boys and 0.50 in girls, and abdominal overweight as >0.48 in boys and >0.47 in girls. Body fat percentage (%BF) was estimated by plicometry using the Siri equation [33], with a previous calculation of density [34,35]. Adiposity levels were classified according to the references for the Spanish youth population [36]. 

#### 2.2.2. CEBQ Questionnaire 

As indicated above, the CEBQ [16], provides information on the response to satiety, taste for food, speed of intake, and emotional food consumption. It is a validated questionnaire with 35 items that assess eight sections of eating behavior and whose questions are answered on a Likert-type scale with an option to score from 0 to 4 according to the intensity of the behavior (where 0 = never, 1 = rarely, 2 = sometimes, 3 = often and 4 = always).

The items are classified into eight subscales: food responsiveness (FR; 5 items), enjoyment of food (EF; 4 items), emotional overeating (EOE; 4 items), desire for drinks (DD; 3 items), slowness in eating (SE; 4 items), satiety responsiveness (SR; 5 items), food fussiness (FF; 6 items) and emotional under-eating (EUE; 4 items). The first four items (FR, EF, EOE and DD), have a positive focus or pro-intake dimension, while the last four (SE, SR, FF and EUE) relate to anti-intake habits. Pro-intake behaviors integrate those habits that favor food consumption, while anti-intake behaviors encompass those habits that lead to avoidance of food consumption. The questions corresponding to each subscale are defined according to the CEBQ’s classification (Table A2). The Spanish version of the CEBQ has been validated [26] and used previously [37].

### 2.3. Statistical Procedures

The internal consistency of the eight subscales of the CEBQ questionnaire and reliability estimates were determined using Cronbach’s alpha. Depending on the normality of the variables, ANOVA, Mann Whitney U tests were performed to compare the mean scores of each subscale of the CEBQ according to nutritional categories. Logistic regression models were applied to establish, as independent variables, the CEBQ subscale score and, as dependent variables, nutritional categories categorized dichotomously according to excess weight, abdominal obesity or high %BF. In these models, sex, age and level of physical activity previously coded according to WHO recommendations were included as covariates [38]. Statistical analysis was performed using R 4.1.2 software. Statistical significance was considered when *p* < 0.05.

## 3. Results

### 3.1. Internal Consistency of the Subscales and Factor Structure of the CEBQ Questionnaire

First, the internal consistency of the CEBQ questionnaire in the present sample was assessed using Cronbach’s Alpha. Internal consistency was adequate (Cronbach’s alpha above 0.7) for all factors except subscales 1 and 8. The unweighted mean factor scores (±SD) and internal reliability estimates (Cronbach’s Alpha) for the CEBQ factors are presented in Table 1.

### 3.2. Sample Characterization

According to BMI, 6.70% of the participants were underweight and 35% had excess weight (24% overweight and 11% obese). Regarding the WHtR, 14.80% were overweight, and 31.80% abdominal obese. According to %BF, 51.20% were classified as having high adiposity (19.40% between 90th–97th percentiles and 31.80% > 97th percentile). Significant differences were found between sexes in the categorization of the sample based on BMI, WHtR and %BF (*p* < 0.001 *), with the male sex having the highest percentage of overweight in all three classifications (Table A3). 

### 3.3. Comparison between Mean Scores of CEBQ Scales and Nutritional Status 

Figure 1, Figure 2 and Figure 3 show a clear trend towards higher scores on the pro-intake subscales and lower scores on the anti-intake subscales as BMI, abdominal obesity, and relative adiposity categories increase. Figure 1 represents separately the trend of the mean scores on the pro-ingestion and anti-ingestion scales, classified according to the nutritional category of each participant according to the body mass index (BMI) categories [30,31]. The trend observed is that the higher the level of overweight, the higher the mean score on the pro-intake scales and the lower the score on the anti-intake scales. Figure 2 represents the trend of the mean scores on the pro-intake and anti-intake scales according to the nutritional category of the sample diagnosed from the waist-to-height ratio (WHtR) [32]. Participants with overweight or abdominal obesity achieved higher mean scores on the pro-intake scales and lower scores on the anti-intake scales. Figure 3 represents the trend of the mean scores on the pro-ingestion and anti-ingestion scales as a function of the nutritional category established on the basis of body fat percentage (%BF) [36]. The general trend observed is that the higher the percentage of body fat, the higher the mean score achieved in the pro-intake scales and the lower in the anti-intake scales.

Table 2 compares the mean scores of the different subscales of the CEBQ as a function of nutritional status as assessed by BMI, WHtR and %BF. In the pro-intake dimension, scores for the subscales EF, FR and EOE were higher (*p* < 0.05) in overweight schoolchildren according to BMI or above the cut-off point for WHR and %BF. The score for the DD subscale was higher only for the abdominal obese. On the other hand, they obtained lower scores (*p* < 0.05) for the SR and SE subscales for the anti-intake dimension than their no obese peers.

As the regression model (Table 3) shows, in general terms, higher mean scores on the pro-intake scales translate into a higher risk of excess weight, abdominal fat, or high %BF. For example, each point scored on the FR and EOE subscales increases the risk of overweight by 2.385 and 2.253 times, respectively. Likewise, each point obtained in the EF subscale increases the likelihood of having high adiposity by 1.8 times. In contrast, the higher the score on the anti-intake subscales (SR and SE), the lower (*p* < 0.05) the risk of being overweight or obese, and the lower the risk of having a high %BF.

## 4. Discussion

Previous research yields results similar to those obtained in our study, showing a significantly lower satiety response capacity in children and adolescents with obesity, as well as a greater enjoyment of food, high responsiveness to external stimuli associated with increased food intake, and a tendency to eat at a faster rate [24,39,40]. Two recently published major studies provide a comprehensive review of eating behaviors linked to childhood obesity, with an emphasis on appetite control and satiety regulation. They have shown that aspects such as satiety responsiveness, responsiveness to food and the tendency to overeat, which are collected in CEBQ, are positively associated with BMI in children [41,42]. Several theories have been put forward to explain delayed satiety in overweight schoolchildren. These include the ability to ingest food without hunger, larger gastric size, metabolic-hormonal dysregulation associated with appetite–satiety control, and greater sensitivity to external factors that predispose to caloric, fatty or sweet products [43]. Similarly, emotional overeating, primarily associated with situations such as anxiety or boredom, or emotional eating due to food restrictions, is associated with an increased risk of developing obesity. On the other hand, several studies suggest that non-hunger eating may be an exciting predictor of weight and obesity at an early age, although the evidence is limited. This is because children who eat more in the absence of hunger are more likely to be able to eat again in a shorter time after a meal, especially more palatable, high-fat and high-calorie foods [44]. 

A sample of 240 Portuguese schoolchildren aged 3–13 years also found a significant association between scores on all pro-intake subscales of the CEBQ and increased risk of elevated BMI. In particular, the risk of obesity was associated with a weaker satiety response and greater food enjoyment [24]. Another study in Portugal involving 2951 schoolchildren concluded that high scores on the pro-intake and low scores on the anti-intake subscales at seven years of age were associated with increased cardiometabolic risk at ten years of age and vice versa [40]. Similar research involving 406 London schoolchildren aged 7–12 years found significant associations between subscales of emotional overeating, increased enjoyment of food, and increased desire to drink with higher adiposity and weight [39]. However, as in the present study, no relationship was observed between EUE score and nutritional status. It is worth noting that some review papers report a close relationship between EOE and emotional disturbances, especially if they are of a negative nature [42]. At the same time, other authors underline an evolutionary tendency to overeat, which generally promotes a higher intake of snacks and low-quality foods [45].

Our results are also consistent with previous findings on the association between lower scores on the anti-intake subscales of the CEBQ in overweight schoolchildren and higher scores in underweight schoolchildren. In particular, a study with a sample of 7295 schoolchildren from the Generation R Study cohort found that children rated by the CEBQ as “more irritable towards food,” less enjoyable, more avoidant, or more likely to be satiated sooner, had significantly lower BMI and %BF [46]. Similarly, a study involving 2500 schoolchildren aged 3–10 years in Bosnia and Herzegovina also found a linear increase in BMI as a function of scores on the pro-intake subscales, except for the desire to drink, and a decrease in BMI as a function of scores on the anti-intake subscales [23]. In general, underweight and normal-weight schoolchildren appear to exhibit certain behavioral traits that protect against the obesogenic environment, while overweight schoolchildren exhibit the opposite traits considered risk factors, supporting the theory of “behavioral susceptibility to obesity” [47]. 

Several lines of research reflect the possibility that overweight children may have been more vulnerable to the obesogenic environment. This means they have been more receptive to advertising and other external stimuli that encourage a higher intake of caloric and unhealthy products. In addition, behavioral patterns predisposing to obesity that begin in childhood may become more pronounced in adolescence and even more so in adulthood [48]. Since interventions to modify eating behavior are more effective at earlier ages, it is of interest to prevent overweight and obesity and to understand the eating behavior of children and adolescents by using validated questionnaires for an individualized approach [49]. 

The present study has some limitations. As indicated in the material and methods section, fieldwork was conducted during the COVID-19 pandemic. Although children attended school and the sports center relatively usually, security measures slowed anthropometric measurements and limited the number of subjects finally included in the study. It was impossible to obtain a sufficient sample size to separate by age group. On the other hand, it is possible that the COVID-19 pandemic had some effect on the eating behavior of schoolchildren. Another issue is that an exclusively anthropometric nutritional diagnosis was performed, assessing both the weight status and the amount and distribution of fat. Moreover, we have tried to associate eating behavior with this physical condition. For this nutritional diagnosis, we did not use blood biochemistry indicators, as this was not the aim of the study. 

It should be noted that the subjects in the sample are school children, and a certain number of them eat part of their meals at school. For this reason, the answers to the test refer exclusively to the eating behavior of the children at home. Finally, as a limitation to be taken into account, we should mention that Cronbach’s alpha, which measures the reliability of internal consistency, is questionable for subscale 8 (FF) of the CEBQ. However, other authors have obtained similar values for this same subscale. Such is the case of Gao et al. [50], who, analyzing a sample of Chinee schoolchildren, estimated a score of 0.49 for this item. 

In the near future, we intend to analyze whether the pro-intake and anti-intake subscales of the CEBQ also show an association with a genetic risk score constructed from a battery of SNPs that we found to be associated with the anthropometric obesity profile in children [51]. We will thus verify whether eating behavior mediates the phenotypic expression of the genetic component of childhood obesity.

## 5. Conclusions

The present study shows the apparent association between anthropometric nutritional status and scores on the subscales of the psychometric test CEBQ. In all pro-intake subscales, schoolchildren with overweight, abdominal obesity or high %BF scored higher. In contrast, in the anti-intake subscales, the average scores were lower than those of their normal-weight peers. This confirms that overweight or obese schoolchildren have a lower satiety response, faster food intake and a pattern of emotional overeating. Given the association between eating behaviors and obesity, it would be essential to know the food-related behavior pattern of the child and adolescent population for a more complete and comprehensive nutritional approach. In this sense, tools such as the CEBQ can be very useful.

## Figures and Tables

**Figure 1 ijerph-20-04186-f001:**
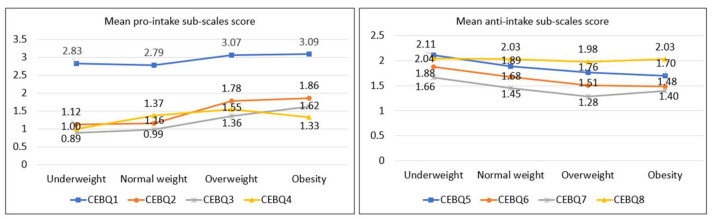
Mean scores of the pro-intake and anti-intake dimensions according to body mass index (BMI) categories [30,31]. Contrast of means using the H-Kruskal Wallis test; *p*-value CEBQ1 = 0.010; *p*-value CBQ2 < 0.001; *p*-value CBQ3 = 0.003; *p*-value CBQ4 = 0.172; *p*-value CEBQ5 = 0.011; *p*-value CEBQ6 = 0.004; *p*-value CEBQ7 = 0.211; *p*-value CEBQ8 = 0.582.

**Figure 2 ijerph-20-04186-f002:**
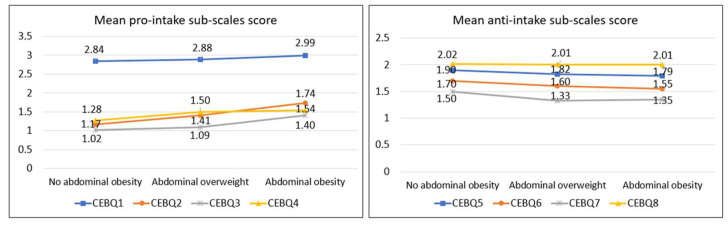
Mean scores of the pro-intake and anti-intake dimensions according to waist-to-height ratio (WHtR) categories [32]. Contrast of means using the H-Kruskal Wallis test; *p*-value CEBQ1 = 0.282; *p*-value CBQ2 < 0.001; *p*-value CBQ3 = 0.048; *p*-value CBQ4 = 0.030; *p*-value CEBQ5 = 0.259; *p*-value CEBQ6 = 0.051; *p*-value CEBQ7 = 0.276; *p*-value CEBQ8 = 0.557.

**Figure 3 ijerph-20-04186-f003:**
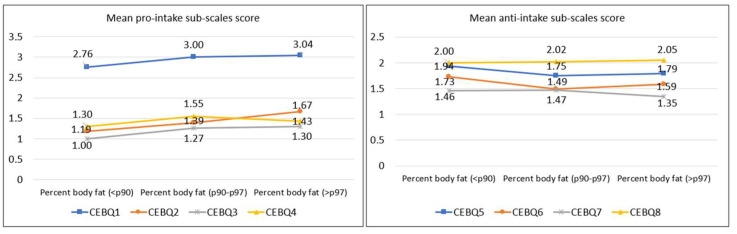
Mean scores of the pro-intake and anti-intake dimensions according to percent body fat (%BF) categories [36]. Contrast of means using the H-Kruskal Wallis test; *p*-value CEBQ1 = 0.005; *p*-value CBQ2 = 0.004; *p*-value CBQ3 = 0.061; *p*-value CBQ4 = 0.143; *p*-value CEBQ5 = 0.020; *p*-value CEBQ6 = 0.014; *p*-value CEBQ7 = 0.743; *p*-value CEBQ8 = 0.779.

**Table 1 ijerph-20-04186-t001:** Mean score and internal consistency of the CEBQ in the analyzed sample. (N = 283).

Dimension	Subscale	Mean (SD)	Cronbach’s Alpha
Pro-intake	1. Enjoyment of food (EF)	2.90 (0.66)	0.631
	2. Food responsiveness (FR)	1.38 (0.99)	0.879
	3. Emotional overeating (EOE)	1.15 (0.91)	0.814
	4. Desire for drinks (DD)	1.39 (0.934)	0.842
Anti-intake	5. Satiety responsiveness (SR)	1.86 (0.52)	0.701
	6. Slowness in eating (SE)	1.64 (0.55)	0.779
	7. Emotional undereating (EUE)	1.43 (0.89)	0.768
	8. Food fussiness (FF)	2.02 (0.39)	0.515

Standard Deviation (SD).

**Table 2 ijerph-20-04186-t002:** Comparison of the mean scores of the CEBQ subscales according to nutritional status assessed by BMI, WHtR and %BF.

Dimension	Subscale	BMI			WHtR			%BF		
		No Excess Weight Median [IQR]	Excess Weight Median [IQR]	*p*-Value	No Excess Abdominal Fat Median [IQR]	Excess Abdominal Fat Median [IQR]	*p*-Value	No Excess Adiposity Median [IQR]	Excess Adiposity Median [IQR]	*p*-Value
Pro-intake	1. Enjoyment of food (EF)	3.00 [2.25–3.25]	3.30 [2.50–3.60]	0.002 *	3.00 [2.50–3.25]	3.00 [2.50–3.60]	0.220	2.75 [2.25–3.25]	3.00 [2.50–3.50]	0.001 *
	2. Food responsiveness (FR)	1.10 [0.40–1.60]	1. 70 [1.00–2.60]	<0.001 *	1.00 [0.40–1.60]	1.40 [0.80–2.40]	<0.001 *	1.00 [0.60–1.60]	1.50 [0.60–2.40]	0.005 *
	3. Emotional overeating (EOE)	1.00 [0.25–1.62]	1.50 [0.75–2.00]	<0.001 *	1.00 [0.25–1.75]	1.25 [0.50–2.00]	0.018 *	1.00 [0.25–1.50]	1.25 [0.50–2.00]	0.014 *
	4. Desire for drinks (DD)	1.33 [0.67–1.83]	1.40 [1.00–2.00]	0.080	1.00 [0.67–1.67]	1.33 [0.67–2.00]	0.015 *	1.00 [0.67–1.67]	1.33 [0.67–2.00]	0.083
Anti-intake	5. Satiety responsiveness (SR)	2.00 [1.60–2.20]	1.80 [1.40–200]	0.011 *	2.00 [1.60–2.20]	1.80 [1.60–2.20]	0.266	2.00 [1.60–2.20]	1.80 [1.60–2.00]	0.018 *
	6. Slowness in eating (SE)	1.75 [1.25–2.00]	1.50 [1.00–1.75]	0.002 *	1.75 [1.25–2.00]	1.50 [1.25–1.75]	0.04 *	1.75 [1.25–2.00]	1.50 [1.25–1.75]	0.008 *
	7. Emotional undereating (EUE)	1.50 [0.75–2.00]	1.25 [0.75–2.00]	0.219	1.50 [1.00–2.19]	1.38 [0.75–2.00]	0.163	1.50 [0.75–2.25]	1.50 [1.75–2.00]	0.580
	8. Food fussiness (FF)	2.00 [1.75–2.25]	2.00 [1.75–2.25]	0.419	2.00 [1.75–2.25]	2.00 [1.75–2.25]	0.613	2.00 [1.75–2.25]	2.00 [1.75–2.00]	0.272

Body Mass Index (BMI); Waist-to-height Ratio (WHtR); Body fat percentage (%BF); Interquartile Range (IQR). Contrast of means with the Man Whitney U test based on BMI, WHtR and %BF. * *p* < 0.05 considered significant.

**Table 3 ijerph-20-04186-t003:** Logistic regression analysis for the association between CEBQ subscales and nutritional status.

Independent Variables	Excess Weight (Overweight or Obesity) by BMI (N = 90)			Excess Abdominal Fat by WHtR (N = 124)			Excess Adiposity by %BF (N = 135)			Subjects with Excess BMI, WtHR, and %BF (N = 78)		
	B	Exp(B)	*p*-Value	B	Exp(B)	*p*-Value	B	Exp(B)	*p*-Value	B	Exp(B)	*p*-Value
CEBQ1	0.489	1.631	0.057	0.120	1.127	0.619	0.566	1.761	0.021 *	0.533	1.703	0.049 *
Female	-	-	-	-	-	-	−0.744	0.475	0.004 **	-	-	-
PA ≥ 420 min/week	-	-	-	-	-	-	-	-	-	-	-	-
CEBQ2	0.869	2.385	<0.001 ***	0.640	1.872	0.010 **	0.416	1.516	0.088	0.797	2.200	0.004 *
Female	-	-	-	-	-	-	−0.745	0.475	0.004 **	-	-	-
PA ≥ 420 min/week	-	-	-	−0.436	0.647	0.096	-	-	-	-	-	-
CEBQ3	0.812	2.253	0.002 **	0.461	1.682	0.035 *	0.531	1.701	0.033 *	0.465	1.592	0.088
Female	-	-	-	-	-	-	−0.733	0.480	0.005 **	-	-	-
PA ≥ 420 min/week	-	-	-	−0.481	0.618	0.067	−0.465	0.628	0.078	-	-	-
CEBQ4	0.490	1.632	0.055	0.543	1.721	0.025 *	0.364	1.439	0.129	0.580	1.786	0.031 *
Female	-	-	-	-	-	-	-	-	-	-	-	-
PA ≥ 420 min/week	-	-	-	−0.460	0.631	0.079	-	-	-	-	-	-
CEBQ5	−0.661	0.516	0.009 **	−0.299	0.742	0.220	−0.630	0.532	0.010 **	−0.616	0.540	0.020 *
Female	-	-	-	-	-	-	−0.744	0.475	0.004 **	-	-	-
PA ≥ 420 min/week	-	-	-	-	-	-	-	-	-	-	-	-
CEBQ6	−0.719	0.487	0.006 **	−0.673	0.510	0.006 **	−0.676	0.509	0.006 **	−0.668	0.513	0.014 *
Female	-	-	-	-	-	-	−0.722	0.486	0.006 **	-	-	-
PA ≥ 420 min/week	-	-	-	−0.446	0.640	0.089	-	-	-	-	-	-
CEBQ7	−0.161	0.851	0.520	−0.188	0.829	0.432	−0.118	0.889	0.622	−0.228	0.796	0.385
Female	-	-	-	-	-	-	-	-	-	-	-	-
PA ≥ 420 min/week	-	-	-	-	-	-	-	-	-	-	-	-
CEBQ8	0.006	1.006	0.982	0.417	1.517	0.084	0.312	1.367	0.193	0.067	1.069	0.813
Female	-	-	-	-	-	-	-	-	-	-	-	-
PA ≥ 420 min/week	-	-	-	-	-	-	-	-	-	-	-	-

Body mass index (BMI); waist-to-height ratio (WHtR); body fat percentage (%BF). Physical activity (PA); CEBQ1. Enjoyment of food (EF); CEBQ2. Food responsiveness (FR); CEBQ3. Emotional overeating (EOE); CEBQ4. Desire for drinks (DD); CEBQ5. Satiety responsiveness (SR); CEBQ6. Slowness in eating (SE); CEBQ7. Emotional undereating (EUE); CEBQ8. Food fussiness (FF); values are represented as β coefficient. * *p* < 0.05; ** *p* < 0.01; *** *p* < 0.001 considered significant.

## Data Availability

The data presented in this study are available on request from the corresponding author. (marrodan@ucm.es).

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
