# Peer review of "Eating Behavior and Obesity in a Sample of Spanish Schoolchildren"

_ijerph, 2023, doi:10.3390/ijerph20054186_

Round 1
Reviewer 1 Report
It is worthy to investigate impact of eating behavior on nutritional status of children because diet and eating behaviors are modifiable factors in prevention of overweight or obesity. This work focused on child nutritional status and eating behaviors. However, some technical flaws could be addressed carefully if a revision will be invited as following:
1. Based on the content of this work, authors focused on obese phenotype and eating behaviors. So title is suggested to be re-considered.
2. A key flaw is unclear statement on features of the sample. Although authors presented 3.2. Sample characterization, they did not present characters of the participants such as number of male, age distribution, physical activities, family background information, et al. I do not know what kind of children were included in this work, which could affect generalization of results. If possible, authors are suggested to add this information.
3. What kind of statistical methods were used in figure 1 and 2. Some statistical testing results should be reported.
4. For table 2, statistical methods used should be noted below the table.
5. For table 2, authors face multiple-comparison actually because of 8 section score of CEBQ, but they seemed not to correct significance level. Pls check it.
6. The analysis in table 3 is very limited because authors did not consider other potential confounders such as sex, age, physical activities, et al, which could impact the nutritional status of children. So the results could be biased.
Author Response
Dear reviewer, we sincerely thank you for your time and comments. We have tried to improve the manuscript according to your recommendations.
We will now respond one by one to your suggestions.
It is worthy to investigate impact of eating behavior on nutritional status of children because diet and eating behaviors are modifiable factors in prevention of overweight or obesity. This work focused on child nutritional status and eating behaviors. However, some technical flaws could be addressed carefully if a revision will be invited as following:
- Based on the content of this work, authors focused on obese phenotype and eating behaviors. So title is suggested to be re-considered. DONE. We have changed in the title "nutritional status" to "obesity".
- A key flaw is unclear statement on features of the sample. Although authors presented 2. Sample characterization, they did not present characters of the participants such as number of male, age distribution, physical activities, family background information, et al. I do not know what kind of children were included in this work, which could affect generalization of results. If possible, authors are suggested to add this information.
DONE. We have added a paragraph in Methodology (Participants) indicating the number and percentage of males and females, expanding the characterization of the sample with data on distribution by age group, maternal and paternal level of education, and hours of physical activity of the participants schoolchildren
- What kind of statistical methods were used in figure 1 and 2. Some statistical testing results should be reported. DONE. A figure caption has been added explaining the statistical test used.
- For table 2, statistical methods used should be noted below the table.
DONE. A table footer has been added to each of the tables explaining the statistical tests used and the relevant explanations in each table
- For table 2, authors face multiple-comparison actually because of 8 section score of CEBQ, but they seemed not to correct significance level. Pls check it.
The CEBQ does not have a total score, but evaluates through 8 different subscales, therefore, the score of each subscale is compared independently. The table shows the values of the statistic and the corresponding p-value for each one of them. The level of significance has been checked.
- The analysis in table 3 is very limited because authors did not consider other potential confounders such as sex, age, physical activities, et al, which could impact the nutritional status of children. So the results could be biased.
The aim is to evaluate how CEBQ is associated with nutritional status. In this regard, it should be borne in mind that each subject has previously been classified according to his or her specific sex and age (Cut-off points for BMI, fat percentage are specific for gender and age).So that, in our opinion, it would be redundant to reintroduce both parameters into the analysis. As for physical activity, as now shown in the characteristics of the participants, this is a relatively homogeneous sample in terms of the number of hours of exercise practiced weekly.

Reviewer 2 Report
1- The Authors need to add to the Title (during CVID 19 Pandemic)
2- Put the symbols in the table and describe them at the end of the table
3- Organize all the tables in terms of alignment and font size
4- Organize the typing in the Appendix Table A1
Author Response
Dear reviewer, we sincerely thank you for your time and comments. We have tried to improve the manuscript according to your recommendations.
We will now respond one by one to your suggestions.
1- The Authors need to add to the Title (during CVID 19 Pandemic)
We have modified the title according to the suggestion of reviewer 1 (replacing nutritional condition with obesity). Although we fully understand yours concern, we have not considered it appropriate to add during the Covid 19 pandemic since in Spain, the pandemic (with the corresponding confinement) was declared in March 2020 and in the present investigation, a good part of the data (60%) were collected in 2019. However, both in the limitations section and in the Methodology (participants) section, the possible effect of this circumstance is indicated.
2- Put the symbols in the table and describe them at the end of the table
DONE. A table footnote has been added to each table explaining the acronyms, symbols, and pertinent information regarding the statistical tests in each case.
3- Organize all the tables in terms of alignment and font size
We have contacted the editor because the alignment and font size of the tables is a layout problem and it is not up to us to modify it. In fact, this format is not the one we originally sent, but the one provided by the magazine.
4- Organize the typing in the Appendix Table A1
DONE. We have aligned the table.

Reviewer 3 Report
The article is well written and identifies an actual problem (the eating behaviour of children age 6-16 years old). The manuscript will be perfect if the authors provide the solutions or the strategy to develop better eating behaviour to children rather than only provide the statistical evidence to the readers.
Author Response
Dear reviewer, we sincerely thank you for your time and kind comments.
The article is well written and identifies an actual problem (the eating behaviour of children age 6-16 years old). The manuscript will be perfect if the authors provide the solutions or the strategy to develop better eating behaviour to children rather than only provide the statistical evidence to the readers.
In response to your suggestion, a consideration has been added to the conclusion

Round 2
Reviewer 1 Report
Thanks for authors addressing my comments. Although they addressed most of my concerns, some issues need to be addressed further as following:
1. I suggest that authors report features of participants which they added in the method in the results with a specific table.
2. For table 2, there is multiple-comparison actually because of 8 section score of CEBQ for each participants, so significant level should be corrected to be 0.05/8, othewise the results could be bised because some section could be not significant.
3. Authors did not consider other potential confounders such as sex, age, physical activities, et al in table 3, which could introuduce some bias. Model with adjusting for main covariates should be used for analysis.
Author Response
Dear Reviewer
We are grateful for your prompt review and favourable comments. We are pleased that our work has improved in your opinion. We will now respond one by one to your latest suggestions.
- Thank you for the authors addressing my comments. Although they addressed most of my concerns, some issues need to be further addressed as follows: I suggest that the authors report the characteristics of the participants they added in the method in the results with a specific table
Done. A table describing the requested characteristics has been added (Table A.1)
- For table 2, there is actually a multiple comparison due to the scoring of 8 sections of the CEBQ for each participant, so the significance level should be corrected to be 0.05/8; otherwise, the results might be bifurcated because some section might not be significant.
Table 2 has yet to be modified, and the reason for this is explained below. This table does not contain multiple comparisons but eight independent comparisons. The CEBQ consists of 8 independent subscales, each comprising several questions that are quantified separately. The CEBQ questions corresponding to each subscale are listed in Appendix Table A2. Consequently, the eight subscales are analyzed separately and have a p-value independent of each other, so the correction suggested in the comments would not apply.
We then refer to two studies published in MDPI journals with a similar methodology. We can see how this significance not needs to be corrected between the eight subscales because they are independent. Therefore, we have followed the same methodology as indicated in the original article by Wardle J et al., 2001, and in all the literature using the CEBQ.
Published articles in which the methodology used in this paper is employed:
- Al-Hamad AH, Al-Naseeb AM, Al-Assaf MS, Al-Obaid SA, Al-Abdulkarim BS, Olszewski PK. Preliminary Exploration of Obesity-Related Eating Behaviour Patterns in a Sample of Saudi Preschoolers Aged 2-6 Years through the Children's Eating Behaviour Questionnaire. Nutrients. 2021 Nov 20;13(11):4156. doi: 10.3390/nu13114156. PMID: 34836411; PMCID: PMC8618833.
Access: https://www.mdpi.com/2072-6643/13/11/4156
(Check: statistical method, and table 5 and 6).
- Ayine P, Selvaraju V, Venkatapoorna CMK, Bao Y, Gaillard P, Geetha T. Eating Behaviors in Relation to Child Weight Status and Maternal Education. Children (Basel). 2021 Jan 7;8(1):32. doi: 10.3390/children8010032. PMID: 33430408; PMCID: PMC7826797.
Access: https://www.mdpi.com/2227-9067/8/1/32
(Check statistical method and table 4).
- The authors did not consider other potential confounders such as gender, age, physical activities, etc. in table 3, which could introduce some bias. A model with adjustment for the main covariates should be used for the analysis. Done
In the first version of our work, we did not consider it appropriate to include the effect of sex and age since each individual had been categorized for the different anthropometric indicators (weight status, abdominal obesity, and level of adiposity) according to these factors.
This procedure is applied or recommended by some authors such as Warketin S et al., who, in the Methodology section, quote, "We describe the crude regression coefficients between the AEBQ subscales and BMIz (unadjusted for sex and age) since BMIz is already age- and sex-specific.".
Warkentin S, Costa A, Oliveira A. Validity of the Adult Eating Behavior Questionnaire and Its Relationship with Parent-Reported Eating Behaviors among Adolescents in Portugal. Nutrients. 2022 Mar 19;14(6):1301. doi: 10.3390/nu14061301. PMID: 35334958; PMCID: PMC8949228.
Access: https://www.mdpi.com/2072-6643/14/6/1301
However, following your suggestion, age, sex, and physical activity have been introduced as covariates in the regression model, modifying table 3. As can now be seen, the results corresponding to the subscales of the CEBQ do not vary notably but allow us to clarify the interaction of these covariates on the phenotype of obesity mediated by eating behavior.
The results in the new table 3 correspond to the most parsimonious logistic regression models for each combination of subscale and phenotype. For this reason, age has been excluded in all analyses, and in some models, the covariates of sex and/or physical activity were not significantly associated.
Once again, we thank you for your time and comments, which have undoubtedly contributed to the quality of our work.